# Effects of Anti-Cancer Drug Sensitivity-Related Genetic Differences on Therapeutic Approaches in Refractory Papillary Thyroid Cancer

**DOI:** 10.3390/ijms23020699

**Published:** 2022-01-09

**Authors:** Hyeok Jun Yun, Minki Kim, Sang Yong Kim, Sungsoon Fang, Yonjung Kim, Hang-Seok Chang, Ho-Jin Chang, Ki Cheong Park

**Affiliations:** 1Gangnam Severance Hospital, Department of Surgery, Yonsei University College of Medicine, 211 Eonjuro, Gangnam-gu, Seoul 135-720, Korea; GSYHJ@yuhs.ac (H.J.Y.); surghsc@yuhs.ac (H.-S.C.); 2Severance Biomedical Science Institute, Graduate school of Medical Science, BK21 Project, Yonsei University College of Medicine, Seoul 03722, Korea; louiskim100@yonsei.ac.kr (M.K.); sfang@yuhs.ac (S.F.); 3Department of Surgery, Yonsei University College of Medicine, 50-1, Yonsei-ro, Seodaemun-gu, Seoul 120-752, Korea; 0101YONG@yuhs.ac; 4EONE-DIAGNOMICS Genome Center, New drug R&D Center, 291 Harmony-ro, Yeonsu-gu, Incheon 22014, Korea; yonjung.kim@edgc.com

**Keywords:** patient-derived papillary thyroid cancer, paclitaxel, sorafenib, lenvatinib, drug-resistant papillary thyroid cancer, EMT, FGFR

## Abstract

Thyroid cancer (TC) includes tumors of follicular cells; it ranges from well differentiated TC (WDTC) with generally favorable prognosis to clinically aggressive poorly differentiated TC (PDTC) and undifferentiated TC (UTC). Papillary thyroid cancer (PTC) is a WDTC and the most common type of thyroid cancer that comprises almost 70–80% of all TC. PTC can present as a solid, cystic, or uneven mass that originates from normal thyroid tissue. Prognosis of PTC is excellent, with an overall 10-year survival rate >90%. However, more than 30% of patients with PTC advance to recurrence or metastasis despite anti-cancer therapy; consequently, systemic therapy is limited, which necessitates expansion of improved clinical approaches. We strived to elucidate genetic distinctions due to patient-derived anti-cancer drug-sensitive or -resistant PTC, which can support in progress novel therapies. Patients with histologically proven PTC were evaluated. PTC cells were gained from drug-sensitive and -resistant patients and were compared using mRNA-Seq. We aimed to assess the in vitro and in vivo synergistic anti-cancer effects of a novel combination therapy in patient-derived refractory PTC. This combination therapy acts synergistically to promote tumor suppression compared with either agent alone. Therefore, genetically altered combination therapy might be a novel therapeutic approach for refractory PTC.

## 1. Introduction

Thyroid cancer (TC) accounts for more than 90% of endocrine cancer, which is the most common endocrine malignancy, and its onset rate has risen over the past four decades [1]. TC is classified into four main types: follicular TC (FTC), medullary TC (MTC), papillary TC (PTC), and anaplastic TC (ATC) [2,3,4]. Furthermore, TC is commonly divided into differentiated and undifferentiated types according to its clinical manifestations [5,6]. Well differentiated TC (WDTC) generally has a favorable prognosis and is relatively easy to treat; however, poorly differentiated TC (PDTC) and undifferentiated TC (UTC) are unusual and aggressive, metastasize early, and have poor prognoses [7,8]. Molecular and biological mechanisms have demonstrated differing clinical behaviors between anti-cancer drug-sensitive PTC and drug-resistant PTC. A number of studies have recommended that the range of mutations may be dissimilar between tumors in anti-cancer drug-sensitive patients and those in drug-resistant patients [9,10,11,12,13,14]. Though a few previous researchers have suggested molecular dissimilarities to describe the aggressive conduct of PTC in drug-resistant patients, no investigation has yet offered an explicit account of the underlying mechanism. In view of the acquired drug resistance in refractory PTC, there is a requirement for useful clinical approaches [15,16,17]. Recent research trends have proven that progress of epithelial–mesenchymal transition (EMT) in refractory cancer cells not only results in metastasis but is also a critical causing factor in drug resistance via fibroblast growth factor receptor (FGFR) signaling [18,19]. While the affinity between EMT and drug resistance was established a short time ago, the system of refractory cancer remains ambiguous. EMT known as the physiological course wherein there is crumble of the cell–cell junctions of epithelial cells, which then change to a condition with properties of migratory cells [20,21]. Even though EMT is a crucial factor in resistance to ErbB aiming composites, deficient perception of the molecular process elementary course has restrained the advance of clinical approaches aimed at drug-resistance condition [22,23]. A few investigations have established that the FGFR signaling pathway plays a decisive role in EMT mediated poor prognosis and drug-resistance of refractory cancer [24,25,26,27] by its alteration of the aggressiveness or cancer stemness of refractory tumor cells.

In this study, we pointed to clarify the association between drug resistant and EMT on refractory PTC to clarify the unfortunate clinical consequences. In particular, we inspected the probable mechanism elementary to anti-cancer drug resistance to promote the expansion of new clinical approaches to address the issue of anti-cancer drug resistace.

## 2. Results

### 2.1. Patient Disease Characteristics

In total, 57 medical records of patients with advanced or metastatic PTC consecutively treated with paclitaxel (*n* = 5), lenvatinib *(n* = 36), or sorafenib (*n* = 16) at our center were analyzed. Patient disease characteristics and demographics by treatment agent are presented in (Figure 1A). The mean age of the patients was 53.2 ± 13.6 years, and 70.2% of patients were female. Distant metastases were observed in 84.2% of patients, and locally advanced PTC was identified in 15.8% of patients. The cases of distant metastatic lesions in the lungs were 40%, 88.9%, and 87.5% in the paclitaxel, lenvatinib, and sorafenib groups, respectively, and in the bone 20%, 19.4%, and 25.0%, respectively. In the lenvatinib group, distant metastatic lesions of the brain were found in 8.3% of patients. All patients had received previous RAI, and the mean cumulative RAI was 442.0 ± 241.3 mCi among those treated with paclitaxel, 553.1 ± 246.1 mCi among those treated with Lenvatinib and 657.5 ± 409.2 mCi among those treated with sorafenib (Figure 1B). Furthermore, 47.4% of patients underwent external beam radiation therapy. Overall survival for 57 patients was 134.3 ± 74.7 months. The patient survival rate was 90.1% at 5 years and 86.1% at 10 years after detection of thyroid cancer (Figure 1C).

### 2.2. Characteristics of Patient-Derived Drug Resistant PTC Cell Lines

Various PTC cell lines were isolated from the specimens obtained from patients (Figure 2A); YUMC-S-P1 and YUMC-S-P2 (first and second isolated patient-derived drug-sensitive PTC), and YUMC-R-P1, YUMC-R-P2, and YUMC-R-P3 (first, second, and third isolated patient-derived drug-resistant PTC) were isolated from patients with PTC who were cured at the Severance Hospital, Yonsei University College of Medicine, Seoul, Republic of Korea. YUMC-R-P1, YUMC-R-P2, and YUMC-R-P3 drug-resistant PTC were more aggressive than YUMC-S-P1 and YUMC-S-P2 drug-sensitive PTC, and metastasis or recurrence was reported in these patients (Figure 2A). Next-generation RNA sequencing to recognize a series of differentially indicated genes revealed that YUMC-R-P1, YUMC-R-P2, and YUMC-R-P3 drug-resistant PTC cells showed induction of the FGFR signaling pathway and EMT markers (*ZEB [zinc finger E-box-binding homeobox]*, *SNAIL [zinc finger protein SNAI1]*, and *TWIST [twist family bHLH transcription factor])* when compared with YUMC-S-P1 and YUMC-S-P2 drug-sensitive PTC cells (Figure 2B). In drug-resistant PTC, the most notably increased genes were fibroblast growth factor *(FGF)*, *FGFR, ZEB*, *SNAIL*, and *TWIST* (Figure 2B). Cancer stem cells (CSCs), FGF, and EMT-related genes were highly expressed in drug-resistant PTC than in drug-sensitive PTC (Figure 2C). Especially, in KEGG pathway analysis, drug-resistant PTC showed upregulated cancer stemness signaling pathway (Wnt, Hedgehog, TGF/SMAD, PI3K/Akt, and PPAR signaling pathway) [28] than drug-sensitive PTC (Figure 2D–F).

Cumulatively, the findings regarding drug-resistant PTC can be of immense importance to the management and therapeutic trials of recurrence and metastasis in patients with aggressive PTC.

### 2.3. Combination of Paclitaxel and Lenvatinib Was More Effective Than Either Agent Alone or a Combination of Paclitaxel and Sorafenib

Based on the results of RNA-Seq-based transcriptome analysis (Figure 2), we hypothesized that the inhibition of FGFR could be a crucial therapeutic approach in paclitaxel- or sorafenib-resistant cancer cells. Consequently, we selected lenvatinib, a multi-receptor tyrosine kinase inhibitor, and evaluated the synergistic anti-cancer effects of its combination with paclitaxel. However, as lenvatinib is included in the signaling pathway of sorafenib as well, a combination of the lenvatinib and sorafenib was excluded. For the assessment of anti-cancer efficacy of paclitaxel, sorafenib, and lenvatinib on patient-derived drug-sensitive and drug-resistant PTC cells, we investigated the proliferation of YUMC-S-P1, YUMC-S-P2, YUMC-R-P1, YUMC-R-P2, and YUMC-R-P3 in the presence of drugs either in combination or alone. Cell viability was suppressed more efficiently with a combination of sorafenib or lenvatinib with paclitaxel than either agent alone and in a dose-dependent manner (Figure 3A,B). Notably, cell viability of drug-sensitive PTC was well-suppressed, regardless of which combination was used and even if the drugs were used alone (Figure 3A). However, cell viability of paclitaxel- or sorafenib-resistant PTC was not meaningfully impacted following treatment with paclitaxel or sorafenib, either separately or in combination (Figure 3B). However, the combination of paclitaxel and sorafenib seemed a little more effective than using either agent alone. While lenvatinib demonstrated more effective suppression than paclitaxel or sorafenib, the synergistic anti-cancer effect of the paclitaxel and lenvatinib combination showed noteworthy suppression of paclitaxel or sorafenib-resistant PTC (Figure 3B). This result proved that FGFR inhibition was critical in FGFR-EMT-mediated drug-resistant PTC. The combination therapy had a half-maximal inhibitory concentration (IC_50_) than that with paclitaxel or sorafenib treatment alone or with lenvatinib in YUMC-R-P1, YUMC-R-P2, and YUMC-R-P3 cells (Figure 3C). These results demonstrated that cotreatment with paclitaxel and lenvatinib might be a new clinical approach worth exploring for targeting drug-resistant PTC.

### 2.4. Paclitaxel and Lenvatinib Cotreatment Induced Nuclear Translocation-Mediated Apoptosis in YUMC-R-P1, YUMC-R-P2, and YUMC-R-P3 Cells

Next, we tested the process of action of the synergistic anti-cancer effects of combined treatment with paclitaxel and lenvatinib in YUMC-R-P1, YUMC-R-P2, and YUMC-R-P3 cells. We used flow cytometry, immunoblot (whole-cell lysate or cellular fractionation), and immunofluorescence analyses with either paclitaxel, lenvatinib, and sorafenib alone or with combined paclitaxel and lenvatinib. The combination of paclitaxel and lenvatinib significantly induced the sub-G_0_/G_1_ population, thus resulting in induction of apoptosis (Figure 4A,C,E) in YUMC-R-P1, YUMC-R-P2, and YUMC-R-P3 cells compared with the use of either agent alone. Immunoblot analysis of the protein expression levels demonstrated that cotreatment with paclitaxel and lenvatinib resulted in a marked induction of C/EBP homologous protein (CHOP) levels, which are active apoptosis markers associated with ER (endoplasmic reticulum) stress, compared with that seen when using either agent alone or when combined paclitaxel and sorafenib was used (Figure 4B,D,F). In contrast, compared with using either agent alone, cotreatment resulted in a decrease in anti-apoptotic factor levels of Bcl-2, ZEB, Snail, p-ERK, MEK, and PKC, which are noted markers of the FGFR signaling pathway and EMT levels (Figure 4B,D,F). These results could indicate that the combination of paclitaxel and lenvatinib is most efficient in suppressing paclitaxel- or sorafenib-resistant PTC. Immunofluorescence analysis indicated that cytochrome *c* was translocated to the nucleus, thus suggesting that cotreatment with paclitaxel and lenvatinib induced apoptosis more significantly than either paclitaxel, sorafenib, or lenvatinib alone via a cytochrome *c*-reliant pathway (Figure 4G–I). Immunoblot analysis after cellular fractionation validated that cytochrome *c* was translocated to the nucleus after cotreatment or treatment with either agent alone (Figure 4J–L). In summary, these results demonstrate that the highest level of apoptosis was induced following cotreatment with paclitaxel and lenvatinib through ER stress- and cytochrome *c*-dependent pathways in patient-derived drug-resistant PTC cells.

### 2.5. Paclitaxel and Lenvatinib Cotreatment Remarkably Suppressed Tumor Growth in a Mouse Xenograft Model with Patient-Derived Drug-Resistant PTC Cell Lines

To determine the synergistic in vivo anti-cancer efficacy of the cotreatment with paclitaxel and lenvatinib, we established a mouse xenograft model with YUMC-R-P1, YUMC-R-P2, and YUMC-R-P3 patient-derived PTC cell lines (Figure 5). We found that paclitaxel, sorafenib, and lenvatinib alone could not significantly inhibit tumor growth; however, compared with using these agents alone, cotreatment with paclitaxel and lenvatinib induced drastic tumor shrinkage (Figure 5A,D,G). Of note, cotreatment with paclitaxel and sorafenib seemed to induce tumor shrinkage compared with that seen using paclitaxel, sorafenib, or lenvatinib treatment alone. However, tumor growth is steadily increasing (Figure 5A,D,G). Mice in the paclitaxel and lenvatinib cotreatment group had a markedly smaller dissected tumor weight than mice treated with paclitaxel, sorafenib, or lenvatinib alone or in combination with paclitaxel and sorafenib (Figure 5B,E,H). There was no evidence of systemic toxicity and mortality in any group. The body weight of mice did not remarkably differ between the groups (Figure 5C,F,I). Anti-apoptotic activity is critical for the estimation of tumorigenesis, and Bcl-2 serves as a marker of anti-apoptotic activity; therefore, we confirmed its presence by performing immunohistochemistry analysis on YUMC-R-P1, YUMC-R-P2, and YUMC-R-P3 cell xenograft tumors. We found that the mice in the paclitaxel and lenvatinib cotreatment group demonstrated the maximum decrease in Bcl-2 expression among all groups (Figure 6A–C).

Cumulatively, the combination of paclitaxel and lenvatinib demonstrated potent anti-cancer effects in a patient-derived EMT-FGFR signaling pathway-mediated drug-resistant PTC cell xenograft model. 

## 3. Discussion

The mortality rate of TC is the highest among common endocrine-mediated malignancies [29], and TC incidence is steadily increasing worldwide, including in the Republic of Korea [29,30,31,32]. TC is projected to become the fourth leading cause of cancer worldwide [31]. Over the past two decades, the universal age-standardized incidence of TC has increased by approximately 25% [31,33,34]. This global increase has been attributed to causes such as increased risk factors, early detection of cancer, increased environmental risk factors, and iodine levels [31]. PTC is known as a general endocrine cancer with excellent prognosis [35,36]. However, unfortunately, several studies have demonstrated that recurrent or metastatic PTC is refractory to most medical treatments [8,37,38,39,40]. Such refractory PTC is usually slothful; however, following anaplasia of the lesion, it becomes undifferentiated cancer that then expands quickly, leading to poor prognosis [41,42]. During TC tumorigenesis, various oncogenic mechanisms as well as cytogenetic events have been reported [43,44,45,46]. In particular, according to well-known previous studies, the ret proto-oncogene rearrangement is a particular genetic alteration observed in PTC but never in UTC [43,47]. The mechanism of aggressiveness in refractory PTC has not been clarified so far. In our next research, we will strive to elucidate the cause for the aggressiveness in refractory PTC. For profiling of the genetic alterations, mRNA-seq analysis was performed to compare patient-derived anti-cancer drug-sensitive PTC and drug-resistant PTC. Countless genes were suggested between drug-sensitive and drug-resistant PTC, revealing that EMT was the principal factor in cancer stemness and aggressiveness [48]. EMT-related genes were more highly induced in drug-resistant PTC when compared with that in drug-sensitive PTC. There has been much debate over the reasonable therapy for patients with drug-sensitive and drug-resistant PTC. We especially concentrated on inspecting cancer cells separated from PTC patients who indicated drug resistance. Established studies have indicated that therapeutic failure in refractory cancer is due to drug resistance [49,50,51]. EMT of cancer is identified as metastasis, invasion, malignant progression, and drug resistance [52,53]. CSCs, discovered in cancer, were hypothesized to be behind the poor prognosis. Furthermore, their stem cell-like oddities [54,55], such as the capability of self-renewal, led them to induce recurrence, metastasis, and therapeutic resistance in tumors [56]. Previous studies have verified a connection among drug resistant, EMT, and CSCs [57,58,59,60]. Particularly, CSCs that demonstrated EMT were regarded to be critical for metastasis and drug resistance, as has been reported in innumerable human drug-refractory cancers [57,58,61]. A few studies have also demonstrated the association between drug resistance and EMT in CSCs [57,58,62]. Of note, in the current study, refractory PTC had properties alike to CSCs in terms of EMT-mediated stemness.

In the current research, the result of the *in vivo* model proved that sorafenib, lenvatinib, or paclitaxel alone could not conspicuously suppress drug-resistant PTC when compared with drug-sensitive PTC. However, sorafenib or lenvatinib with paclitaxel markedly suppressed the tumors in drug-resistant PTC. Additionally, selected drug combinations based on genetic alterations were demonstrated to be the most efficient at tumor suppression. Our findings highlight that paclitaxel, sorafenib, or lenvatinib-resistant patient-derived PTC cells could be frustrated by synergistic effects of combined sorafenib or lenvatinib with paclitaxel via the inhibition of Bcl-2 and EMT-mediated FGFR signaling pathway. It is an established fact that the proto-oncogene and anti-apoptotic factor Bcl-2 plays a major role in the regulation of apoptosis [63,64]. These could prove that Bcl-2 is associated with EMT, a crucial mechanism in the development of drug resistance in PTC [65]. Furthermore, release of cytochrome *c* from the mitochondria is a crucial event for apoptosis [66]. Particularly, the released cytochrome *c* induces the caspase-mediated apoptotic signaling pathway [67]. We proved that cytochrome *c*, released from the mitochondria, induced apoptosis and gradually amassed in the nucleus, as proven by both immunofluorescence and cellular fractionation after treatment with sorafenib or lenvatinib with paclitaxel. These findings might be valuable in designing future rational clinical approaches for patients with refractory PTC. Our future and prospective treatment strategies are to find new mechanisms or hit-to-lead for refractory cancer based on mRNAseq of patient derived cancer cells. In particular, we will focus on ‘target validation and selection’, ‘hit and lead generation’, ‘lead optimization to identify a clinical drug candidate’, and ‘biomarker-led clinical trials’. Other investigations include enlargement of the envelope of druggability for less manageable targets, surmounting and understanding drug resistance, and designing intelligent and available drug combinations. Of course, further investigations are needed to assess the current clinical approach. Furthermore, additional studies are needed due to the fact that only a few patient outcomes were assessed in the current study. 

Nevertheless, it is worth noting that these discoveries recommend that therapeutic approaches were founded on genetics, for instance, those involving downregulation of the EMT-mediated FGFR signaling pathway, are possible novel clinical approaches for patients with refractory cancer with drug-resistant properties.

## 4. Materials and Methods

### 4.1. Study Design and Ethical Considerations

This study was a retroactive, single center examination of patients diagnosed with PTC (January 2003–December 2019), as detailed in our previous study [68]. All courses entailing patients were achieved in proportion to the institutional ethical standards, whole applicable national/local regulations, and guidelines of the 1964 Helsinki Declaration and its later amendments. The study proceedures were authorized by the Institutional Review Board (IRB) of Severance Hospital, Yonsei University College of Medicine (IRB protocol: 3-2019-0281). Cancer cells were obtained from fresh tissue of patients at the Severance Hospital, Yonsei University College of Medicine, Seoul, Korea.

### 4.2. Patients

#### 4.2.1. Patient 1 

YUMC-R-P1 was a 52-year-old woman with PTC. This patient had multiple tumors and extensive extrathyroidal extension. This woman experienced a bilateral total thyroidectomy with main compartment neck resection. One year after surgery, metastasis to the mediastinum and right lateral cervical lymph nodes was confirmed, and she underwent a mediastinal dissection through partial sternotomy and right modified radical neck dissection. The specimens for culture were gained after the last surgery. After surgery, the pathology state revealed the existence of metastatic papillary thyroid carcinoma. 

#### 4.2.2. Patient 2 

YUMC-R-P2 was a 57-year-old man with poorly differentiated thyroid carcinoma. After a bilateral total thyroidectomy with main compartment neck resection, this man experienced left radical nephrectomy and right lung wedge resection for kidney and lung metastasis. Afterward, he underwent a right modified radical neck dissection, left lateral selective lymph node dissection, and 2 regional lymph node dissections. The specimens for culture were gained after the last surgery. After regional lymph node dissection (left level III), the pathology state revealed the existence of metastatic poorly differentiated thyroid carcinoma. 

#### 4.2.3. Patient 3

YUMC-R-P3 was a 34-year-old woman with PTC. She experienced a bilateral total thyroidectomy with main compartment neck resection and right modified radical neck dissection. One year after surgery, metastasis to the mediastinum was confirmed, and she underwent mediastinal dissection through transcervical approach. Then, metastasis to the upper mediastinum was confirmed, and mediastinal dissection was additionally performed. The specimens for culture were obtained after the second surgery. After surgery, the pathology state revealed the existence of metastatic papillary thyroid carcinoma. 

### 4.3. Specimens of Patients

Fresh tumors were gained from patients with biochemical and histologically established PTC who were attended at the Severance Hospital, Yonsei University College of Medicine, Seoul, Korea. Fresh tumors were acquired via surgical resection of PTC metastatic and primary sites. 

### 4.4. Tumor Cell Isolation and Primary Culture

After resection, tumors were placed in phosphate-buffered saline (PBS) with antibiotics and antifungal and moved to the laboratory. Further details on the protocol can be found in our previous article [68].

### 4.5. mRNA-Seq Data

We preprocessed the raw reads from the sequencer to remove low quality and adapter sequence before analysis and aligned the processed reads to the Homo sapiens (GRCh37) using HISAT v2.1.0 (KIM et al., 2015) as detailed in previous research [69].

### 4.6. Statistical Aanalysis of Gene Expression Level

The relative abundances of gene were measured in Read Count using StringTie. Further details on the protocol can be found in ‘www.frontiersin.org’(accessed on 15 December 2021).

### 4.7. Hierarchical Clustering

Hierarchical clustering analysis was carried out with complete linkage and Euclidean distance as a measure of similarity to indicate the expression patterns of differentially expressed transcripts which are satisfied with |fold change|≥2 and independent t-test raw *p* < 0.05. All data analysis and visualization of differentially expressed genes was conducted using R 3.5.1 (www.r-project.org, accessed on 15 December 2021).

### 4.8. Cell Culture

The PTC cell lines YUMC-R-P1, -P2, and -P3 were obtained by tumor cell isolation from the patients and cultured in RPMI-1640 medium with 15% fetal bovine serum (FBS; authenticated by short tandem repeat profiling/karyotyping/isoenzyme analysis).

### 4.9. Cell Viability Assay

Cell viability was measured using the MTT (3-(4,5-Dimethylthiazol-2-yl)-2,5-Diphenyltetrazolium Bromide) assay, cells were seeded in 96-well plates at 7 × 10^3^ cells per well and incubated overnight to achieve 80% confluency. Further details on the protocol can be found in our previous article [68]. Data were indicated as a percentage of the signal observed in vehicle-treated cells and are shown as the means ± SEM of triplicate experiments.

### 4.10. Cell CycleAnalysis Using Flow Cytometry

Cells were treated paclitaxel, sorafenib, and lenvatinib either alone or in combination (except a combination of sorafenib and lenvatinib) in RPMI-1640 medium containing 15% FBS for 40 h. Further details on the protocol can be found in our previous article [68]. This experiment was repeated in triplicate and the results were averaged.

### 4.11. Immunofluorescence Analysis and Confocal Imaging

The expression of cytochrome *c* was analyzed by immunofluorescence staining. Cells grown on glass-bottomed dishes (MatTek, Ashland, MA, USA) were fixed with 4% formaldehyde solution (R&D Systems, Abingdon, UK) for 10 min and permeabilized with 0.5% TritonX-100 in phosphate-buffered saline (PBS) for 10 min. Slides were air-dried, washed with PBS, and incubated with anti-cytochrome *c* (1:25; Abcam, Cambridge, UK) in 3% bovine serum albumin in PBS. Further details on the protocol can be found in our previous article [68]. Images were observed under a confocal microscope (LSM Meta 700; Zeiss, Oberkochen, Germany) and were analyzed using Zeiss LSM Image Browser, version 4.2.0121.

### 4.12. Cellular Fractionation

Cellular fractions were contrived with the NEPER Nuclear and Cytoplasmic Extraction kit (Thermo Scientific, 78833, Waltham, MA, USA) in conformity with the manufacturer’s dicates.

### 4.13. Immunoblot Analysis

After transfer process, the primary antibodies against CHOP purchased from Cell Signaling Technology; Danvers, MA, USA, Bcl-2, cytochrome *c*, Snail1, Zeb1, and histone H2B purchased from Abcam; and PKC, MEK, p-ERK, ERK, and β-actin purchased from Santa Cruz Biotechnology, Santa Cruz, CA, USA were maintained overnight at 4 °C. As detailed in previous research [70]. 

### 4.14. Human PTC Cell Xenograft

Whole experiments were confirmed by the Animal Experiment Committee of Yonsei University. YUMC-R-P1, -P2, and -P3 patient-derived PTC cells (4.4 × 10^6^ cells/mouse) were cultured in vitro, then injected subcutaneously into the upper left flank region of female NOD/Shi-scid, IL-2Rγ KOJic (NOG) mice. After 15 days, tumor cell planting mice were arranged arbitrarily (*n* = 10 per group). They were administered with 25 mg/kg Paclitaxel (i.p.), 80 mg/kg sorafenib (p.o.), and 10 mg/kg lenvatinib (p.o.) either alone or in combination. Tumor volume was quantified every two days by calipers. Tumor volume was gauged by formula: L × S2/2 (L, longest diameter; S, shortest diameter). Mice were kept under specific pathogen-free conditions. Whole experiments were established by the Animal Experiment Committee of Yonsei University.

### 4.15. Immunohistochemistry

Primary monoclonal antibodies against Bcl-2 (Abcam) were diluted with PBS (1:100) overnight at 4 °C. All tissue sections were counterstained with hematoxylin, dehydrated, and mounted.

### 4.16. Image Analysis

The MetaMorph 4.6 software (Molecular Devices, San Jose, CA, USA) was used for computerized quantification of immunostained target proteins.

### 4.17. Statistical Analysis

For the assay of patient report, unequivocal fluctuations were expressed as frequency and proportion, whereas summary statistics (median, range) were used to report continuous data. Survival curves were generated using the Kaplan–Meier method based on the log-rank test. As this was a retrospective analysis, no formal statistical comparisons were performed. Statistical analyses were performed using GraphPad Prism 6.0 software (GraphPad Software, La Jolla, CA, USA), Microsoft Excel (Microsoft Corp, Redmond, WA, USA), and R version 2.17. One-way ANOVA was performed for the multi group analysis, and a two-tailed Student t-test was performed for the two-group analysis. Immunohistochemistry results were subjected to one-way analysis of variance, followed by Bonferroni post hoc test. Values were expressed as the mean ± standard error of mean. *p* values < 0.05 were considered statistically significant.

## 5. Conclusions

The synergistic anti-cancer activity of paclitaxel and lenvatinib was more effective than that of either agent alone in patient-derived drug-resistant PTC. These findings might be useful in designing future clinical studies and aid in the development of effective therapies for patients with advanced or metastatic PTC.

## Figures and Tables

**Figure 1 ijms-23-00699-f001:**
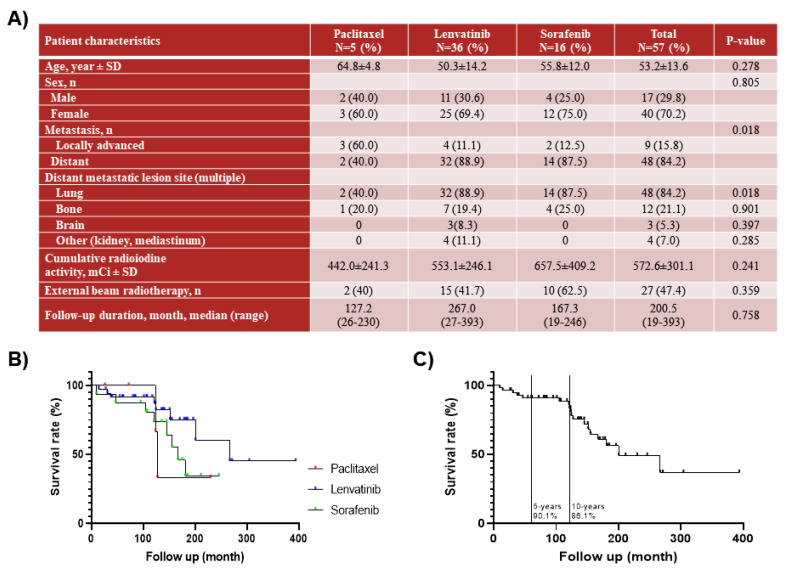
Information on patients with papillary thyroid cancer (PTC) after treatment of paclitaxel, lenvatinib and sorafenib: (**A**) patient characteristics and clinical features, (**B**) overall survival rate after treatment of paclitaxlel, Lenvatinib, and sorafenib, and (**C**) overall survival rate of patients with refractory PTC.

**Figure 2 ijms-23-00699-f002:**
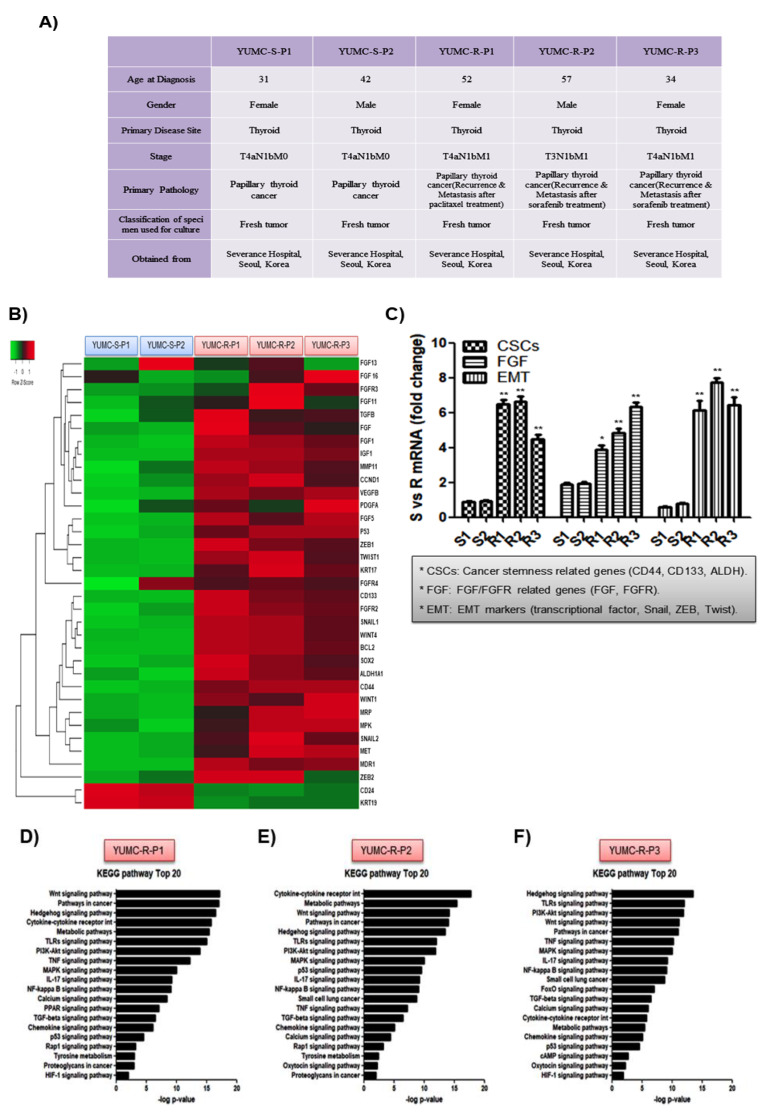
Characteristics of all examined papillary thyroid cancer (PTC) cell lines. (**A**) Characteristics of patient-derived subtypes of PTC cell lines, (**B**) hierarchical clustering of annotated genes revealing distinct gene expression; gene expression profile differences between patient-derived drug-sensitive and drug-resistant PTC cells, and (**C**) gene expression level analysis based on mRNA seq for markers of cancer stem cells (CSC), fibroblast growth factor (FGF)/FGF receptor, and epithelial-mesenchymal transition (EMT) between patient-derived anti-cancer drug sensitive and resistant PTC cells. (**D**,**E**,**F**), bar plot showing 20 significantly enriched upregulated pathways in patient derived anti-cancer drug resistant PTC cells, YUMC-R-P1 (**D**), YUMC-R-P2 (**E**), and YUMC-R-P3 (**F**). * *p* < 0.05 vs. anti-cancer drug-sensitive PTC cells, YUMC-S-P1, ** *p* < 0.01 vs. anti-cancer drug-sensitive PTC cells, YUMC-S-P1.

**Figure 3 ijms-23-00699-f003:**
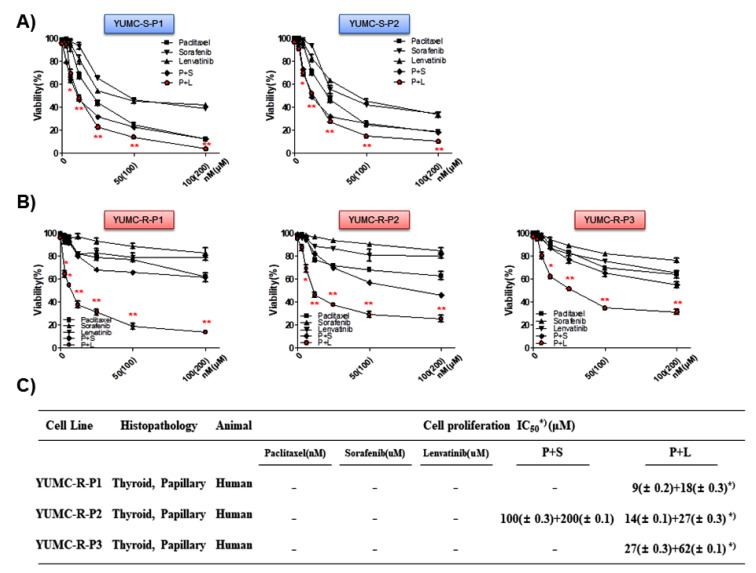
Synergistic anti-cancer effect of paclitaxel and lenvatinib in anti-cancer drug-sensitive and drug-resistant papillary thyroid cancer (PTC) cells. Cell viability of anti-cancer drug-sensitive (**A**, YUMC-S-P1 and YUMC-S-P2) and -resistant (**B**, YUMC-R-P1, YUMC-R-P2, and YUMC-R-P3) PTC cells with paclitxel, sorafenib, and levatinib combined or with each agent alone. Points indicate the mean percentage of the values observed in solvent-treated control. All experiments were performed at least thrice. Data represent mean ± standard deviation. * *p* < 0.05 and ** *p* < 0.01 versus control. (**C**), Half-maximal inhibitory concentration (IC_50_) values for the combination of sorafenib and lenvatinib in anti-cancer drug-sensitive and -resistant PTC cells. Each data point signifies the mean of three independent MTT assays, performed in triplicate. SEM, standard error of the mean; MTT, 3-(4,5-dimethylthiazol-2-yl)-2,5-diphenyltetrazolium bromide. The asterisk indicates the lowest half-maximal inhibitory concentration.

**Figure 4 ijms-23-00699-f004:**
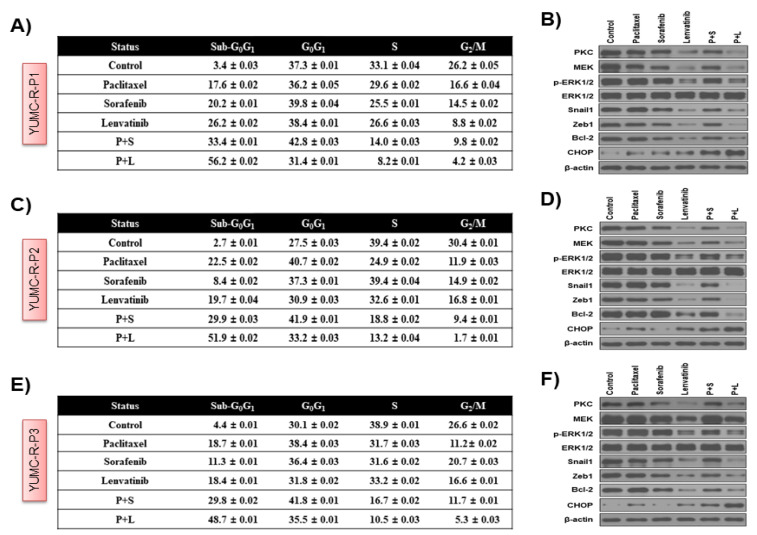
Apoptosis analysis by quantitation of DNA content with propidium iodide, immunoblot, and immunofluorescence analyses. Flow cytometry (**A**, **C**, and **E**) and immunoblot analyses (**B**, **D**, and **F**) of patient-derived anti-cancer drug-resistant papillary thyroid cancer (PTC) cells, YUMC-R-P1, YUMC-R-P2, and YUMC-R-P3. (**A**, **C**, and **E**), Cells were exposed to the indicated inhibitors, harvested, and stained with propidium iodide before analysis by flow cytometry and further assessment using FlowJo version 8. (**B**, **D** and **F**), Immunoblot analysis of the markers of endoplasmic reticulum stress, EMT, and anti-apoptosis in YUMC-R-P1, YUMC-R-P2, and YUMC-R-P3. (**G**, **H**, and **I**), Immunofluorescence examined at 400 × magnification; scale bar: 20 μm and (**J**, **K,** and **L**) subcellular fractionation analysis. (**J**, **K,** and **L**), Cytochrome *c* was the most translocated, and it accumulated in the nucleus in the paclitxel and lenvatinib cotreatment group. Combination of paclitaxel and lenvatinib resulted in the most induced nuclear translocation of cytochrome *c*-mediated apoptosis in anti-cancer drug-resistant PTC cell lines, YUMC-R-P1, YUMC-R-P2, and YUMC-R-P3.

**Figure 5 ijms-23-00699-f005:**
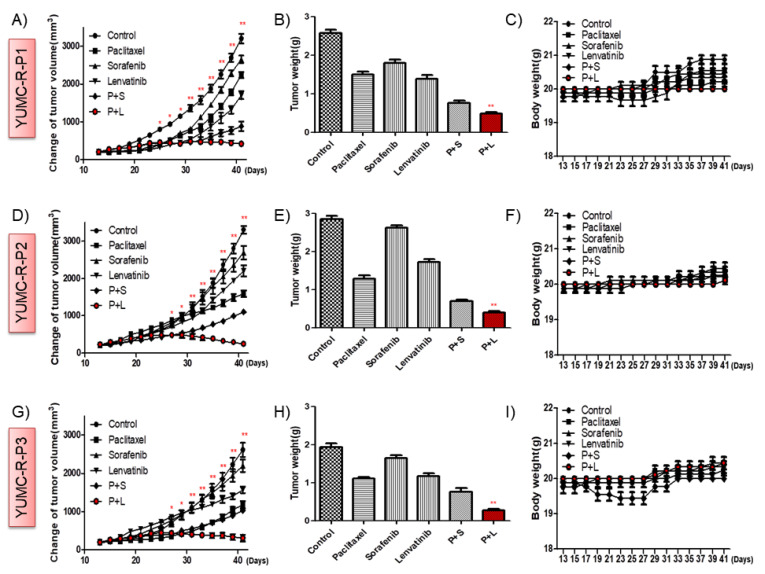
Combination of paclitaxel and lenvatinib produced the highest tumor shrinkage in patient-derived anti-cancer drug (paclitaxel or sorafenib)-resistant papillary thyroid cancer (PTC) cells, YUMC-R-P1, YUMC-R-P2, and YUMC-R-P3 xenografts in vivo. (**A**, **D**, and **G**), Change in tumor volume, (**B**, **E**, and **H**), the dissected tumor weight, and (**C**, **F**, and **I**), change in whole body weight. Paclitxel, sorafenib, and lenvatinib alone had no significant effect on the mouse body weight. * *p* < 0.05 and ** *p* < 0.01, compared with control.

**Figure 6 ijms-23-00699-f006:**
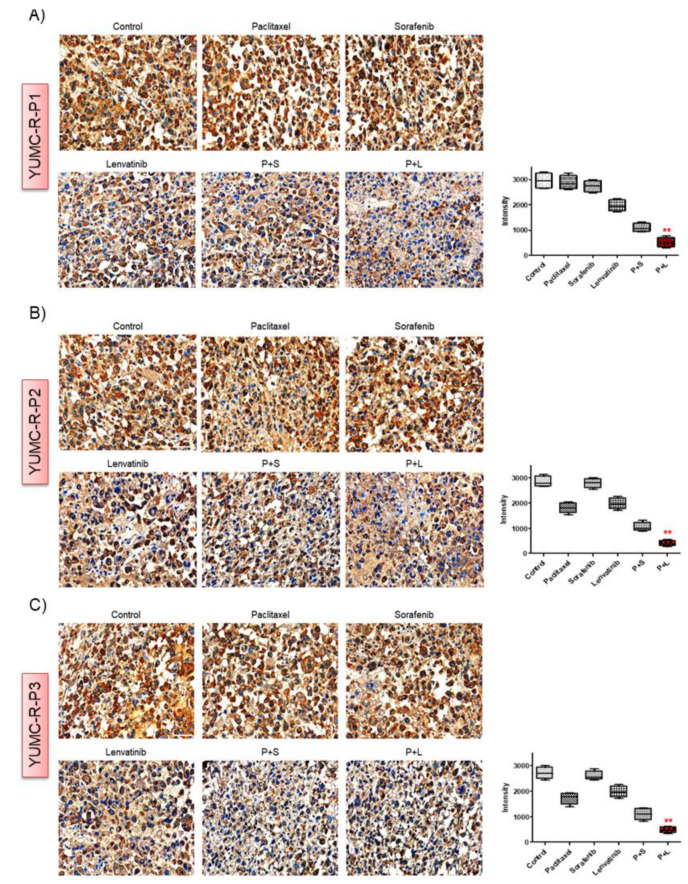
Immunohistochemical analysis of Bcl-2 proteins in tumor tissues following the indicated treatments. (**A**), YUMC-R-P1, (**B**), YUMC-R-P2, and (**C**), YUMC-R-P3 cells examined at 400 × magnification; scale bar: 80 μm. Each assay was performed in triplicate, and the representative images are presented. ** *p* < 0.01 versus control. MetaMorph 4.6 image analysis software was used to quantify the immunostained target proteins.

## Data Availability

The data presented in this study are available on reasonable request from the corresponding author.

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
