# Peer review of "Effects of Anti-Cancer Drug Sensitivity-Related Genetic Differences on Therapeutic Approaches in Refractory Papillary Thyroid Cancer"

_ijms, 2022, doi:10.3390/ijms23020699_

Round 1
Reviewer 1 Report
I found this paper interesting, only few comments are addressed:
- Some limitations should be more carefully discussed.
- Future and prospective treatment strategies should be briefly mentioned based on the authors' experience.
- Please carefully read paper and correct minor language mistakes
Author Response
Reviewer 1
I found this paper interesting, only few comments are addressed:
Some limitations should be more carefully discussed.
Future and prospective treatment strategies should be briefly mentioned based on the authors' experience.
> Reply: I don't know how to thank you enough for reviewing our manuscript. I agree with you completely and follow your professional opinion. I have made the suggested correction. Added sentences were indicated blue color in section of ‘discussion’.
Please carefully read paper and correct minor language mistakes
> Reply: Thank you for your comment. I have made this correction. Thank you again for your review. I hope you are always healthy and happy!!
Reviewer 2 Report
The study by Hyeok Jun Yun and MinKi Kim et al is an interesting attempt to study the effects of combination of three anticancer drugs on PTC samples. The study is interesting and well written, however some questions need to be addressed before considering the work for publication:
- The conclusions show beneficial effect of combination of the three drugs, but were there any adverse effects of the combination of drugs?
- Patients 1 and 3 have been described as “patients with PTC”. Is it possible to further characterize the subtypes of those cases?
Line 335 – Buttumor – a spelling mistake
Line 372 – n – a spelling mistake
Line 377 - Therefore, in this research, we strived to elucidate the cause for the aggressiveness in refractory PTC – this question seems to be not addressed in the manuscript
Line 381 - was reprogrammed reliance on cancer stemness – please rewrite the sentence, so that the meaning is more clear
Author Response
Reviewer 2
The study by Hyeok Jun Yun and MinKi Kim et al is an interesting attempt to study the effects of combination of three anticancer drugs on PTC samples. The study is interesting and well written, however some questions need to be addressed before considering the work for publication:
> Reply: I don't know how to thank you enough for reviewing our manuscript. I agree with you completely and follow your professional opinion. I have made the suggested correction. Corrected sentence were indicated blue color. Thank you again for your review. I hope you are always healthy and happy!!
- The conclusions show beneficial effect of combination of the three drugs, but were there any adverse effects of the combination of drugs?
> Reply: Thank you for this comment. Based on, kidney, spleen and hepatic injury did not shown in xenograft models, combination of the three drugs might be no adverse effects.
- Patients 1 and 3 have been described as “patients with PTC”. Is it possible to further characterize the subtypes of those cases?
> Reply: Both patients had conventional subtypes. There were no specific features except extrathyroidal extension in a specimen from 2 patients and lymphatic involvement in a sample from patient 3. The association between aggressive subtypes (e.g., diffuse sclerosis, solid, tall cell variants) and refractory PTC has not been established but will be investigated in future studies.
- Line 335 – Buttumor – a spelling mistake
> Reply: Thank you for your comment. I have made this correction.
- Line 372 – n – a spelling mistake
> Reply: Thank you for your comment. I have made this correction.
- Line 377 - Therefore, in this research, we strived to elucidate the cause for the aggressiveness in refractory PTC – this question seems to be not addressed in the manuscript
> Reply: Thank you for your comment. I corrected to ‘in this research’ > ‘Our next research’.
- Line 381 - was reprogrammed reliance on cancer stemness – please rewrite the sentence, so that the meaning is more clear
> Reply: Thank you for your comment. I have made this correction.
